# Synergistic Neuromorphic Federated Learning with ANN-SNN Conversion For Privacy Protection

## Abstract

Federated Learning (FL) has been widely researched for the growing public data privacy issues, where only model parameters, instead of private data, are communicated. However, recent studies debunk the privacy protection of FL, showing that private data can be leaked from the communicated gradients or parameter updates. In this paper, we propose a framework called Synergistic Neuromorphic Federated Learning (SNFL) that enhances privacy during FL. Before uploading the updates of the client model, SNFL first converts clients' Artificial Neural Networks (ANNs) to Spiking Neural Networks (SNNs) via calibration algorithms. In a way that not only loses almost no accuracy but also encrypts the client model's parameters, SNFL manages to obtain a more performant model with high privacy. After the aggregation of various SNNs parameters, the server distributes the parameters back to the clients. This design offers a smooth convergence to continue the model training under the ANN architecture. The proposed framework is demonstrated to be private, introducing a lightweight overhead as well as yielding prominent performance boosts. Extensive experiments with different kinds of datasets have demonstrated the efficacy and the practicability of our method. In most of our experimental IID and not extreme Non-IID scenarios, the SNFL technique has significantly enhanced the model performance. For instance, SNFL improves the accuracy of FedAvg on Tiny-ImageNet by 13.79%. Besides, the original image cannot be reconstructed after 280 iterations of attacks with the SNFL method, whereas it can be reconstructed after just 70 iterations with FedAvg.

## 1 Introduction

Recent advancements in machine learning, particularly deep learning, rely heavily on large data sets to obtain decent inference performance. Due to the growing demand for data, it is now necessary to feed models with information from multiple entities. However, this transfer, exchange, and trade of data among entities may violate the General Data Protection Regulation (GDPR) and get punished by the Act (Wachter, 2018), posing an unprecedented challenge to the field of machine learning. Federated learning (McMahan et al., 2017) then emerges and flourishes as a privacy-preserving approach by training a shared model collaboratively while keeping data locally. Despite that the data are stored locally, clients that join the federated learning need to transmit their local gradients to the server to update the shared model. Recent studies Zhu & Han (2020); Zhao et al. (2020); Huang et al. (2021) have revealed that sensitive local data could be leaked from these transmitted local gradients via model inversion attack Zhu & Han (2020). To defend against such kind of attack and prevent privacy leakage, defense strategies including differential privacy (Geyer et al., 2017), secure multi-party computation (Byrd & Polychroniadou, 2020), and MixUp (Zhang et al., 2017) have been developed. In exchange for privacy, the cost is then either severe computational overheads (Hardy et al., 2017) or unavoidable accuracy losses (Kim et al., 2021).

What's the intrinsic source of privacy in these defense strategies? If we consider this question from an information theory perspective, it is indeed the asymmetry of entropy in the encryption and decryption steps for clients and servers when partial encryption information is kept locally only. From this standpoint, as long as an encryption method is capable of inevitability between clients and servers while still allowing for effective aggregation, it would be feasible to improve the privacy

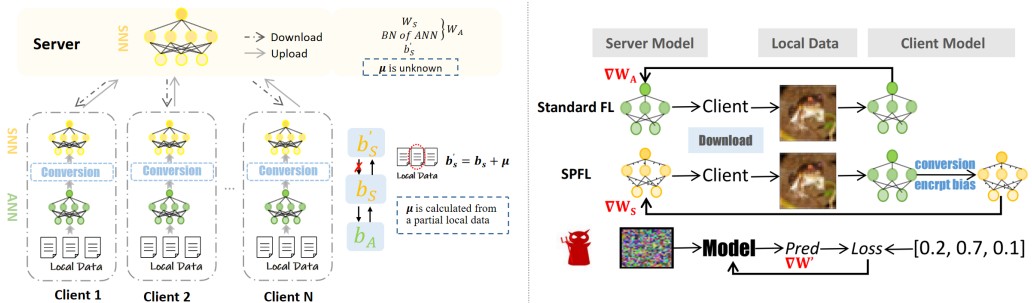

Figure 1: *Right*: Workflow of Synergistic Neuromorphic Federated Learning with ANN-SNN Conversion (SNFL). *Left*: Each client communicates parameters $\nabla W_A$ generated by the model trained on private local data. The attacker updates randomized dummy input and label to minimize the gradient distance $||\nabla W_A - \nabla W'||$. When the optimization is complete, the attacker can obtain the training set from the client. However, in SNFL, the client's model has been converted and calibrated to SNN before communication.

for federated learning. Recent progression in neuromorphic computing, especially the conversion from traditional artificial neural networks (ANNs) to spiking neural networks (SNNs) (Deng & Gu, 2020), provides a pair of source ANN and target SNN that both achieve high accuracy, with the source ANN not recoverable from the resulted SNN (Li et al., 2021b). This property fits naturally with the demand for privacy protection in federated learning. Indeed, if we train ANNs on clients and only send the converted SNNs with partial parameters to the server for aggregation, we can then expect to obtain a feasible privacy-protected FL algorithm with an effective parameter-sharing paradigm. In addition, such an ANN-SNN conversion is lightweight and performance-preserving (or even performance-improving) by careful design. Fig. 1 illustrates the pipeline of our proposed method.

Besides the natural feasibility of SNNs (Esser et al., 2016; Kim et al., 2019), this synergistic framework also brings two additional benefits that are special for federated learning. First, in contrast to existing noise injection methods (*e.g.*, differential privacy Geyer et al. (2017)), our ANN-SNN conversion process is optimized to improve performance by fine-tuning SNN's weights rather than trading off performance drop versus noise level. As a result, our method is able to achieve even better performance against standard federated learning. Second, the SNN emits discrete spikes and is not differentiable, thus the induced synergistic FL could be more robust to small perturbations and adversarial attacks like white-box attacks (Liang et al., 2021). Our contributions are summarized as follows:

- Innovation/Privacy: We design a federated learning framework where the server and clients run two different models in a privacy-preserving manner as a new solution. To the best of our knowledge, our work is one of the first to train different types of neural network models on server and clients.

- Accuracy: Compared to the conventional approach, extensive experiments validate SNFL can deliver similar or superior accuracy relative to other common methods.

- Effectiveness: Based on the SNFL framework, we analyze the backdoor attack and develop a method to simply detect it through abnormal SNN thresholds.

## 2 RELATED WORK

### 2.1 FEDERATED LEARNING (FL)

In federated learning, each client computes a model update, i.e. gradient, on its local data. While sharing gradients was assumed to leak little information about the client's private data, recent papers (Zhu & Han, 2020; Zhao et al., 2020; Huang et al., 2021) devised "gradient inversion attack" in which an attacker listening to one client's communications with the server can begin to reconstruct the client's private data. To defend against this, methods such as gradient clipping (Sun et al., 2019),

perturbing gradients (Zhu & Han, 2020), and robust aggregation (Blanchard et al., 2017; Goryczka & Xiong, 2015; Yin et al., 2018) are commonly used.

## 2.2 SPIKING NEURAL NETWORK (SNN)

Conventionally, there are two distinct routes to obtain a deep SNN (Deng et al., 2020):(1) direct training SNN from scratch, and (2) converting a pretrained ANN to SNN. In this work, we mainly focus on the conversion-based method. ANN-SNN conversion directly reuses the features learned in ANN to obtain a performant SNN. However, it requires a trade-off between inference latency and task performance. Data-based normalization (Diehl et al., 2016) and threshold balancing (Sengupta et al., 2018) are the basic methods of ANN-to-SNN conversion. Then Rueckauer et al. (2016); Han et al. (2020) propose the soft mechanism to reduce information loss by membrane potential reset. Recently, Deng & Gu (2020) analyze the conversion error and propose a shift method to reduce it by half. Li et al. (2020) propose a light pipeline and an advanced pipeline, which apply layer-wise calibration algorithms to modify the network parameters to diminish the conversion error, significantly reducing the required simulation length. We adopt the layer-wise calibration algorithm for high-performance and low-latency SNN in the SNFL framework.

## 3 PRELIMINARIES

In this section, we briefly introduce the concept and the baseline method for Federated Learning (FL). We also point out the privacy issue in FL, which can lead to the leakage of user data.

**Federated Learning (FL).** FL enables mobile devices to collaboratively learn a shared prediction model while keeping all the training data on device, decoupling the ability to do machine learning from the need to store the data in the cloud. Formally, assuming we have $K$ clients, the optimization objective in FL is to solve the following empirical risk minimization problem:

$$\min_{\theta} L(\theta) = \frac{1}{K} \sum_{i=1}^{K} \ell_i(\theta), \text{ where } \ell_i(\theta) = \ell_{(\boldsymbol{x}_i, \boldsymbol{y}_i) \sim \mathcal{D}_i}(\boldsymbol{x}_i, \boldsymbol{y}_i; \theta). \tag{1}$$

Here, $\theta$ is the model parameters vector and $\ell_i(\theta)$ denotes the loss function of $\theta$ evaluated on the $i$-th client's dataset $\mathcal{D}_i$. $(\boldsymbol{x}_i, \boldsymbol{y}_i)$ is the input-label pair on $\mathcal{D}_i$. The goal of this objective is to achieve minimum average loss on each client. We assume $\{\mathcal{D}_i\}_{i=1}^{K}$ are randomly sampled from $\mathcal{D}_{\text{train}}$, where both IID sampling and Non-IID sampling are considered in this work.

To ensure the minimization of average loss on all clients while not sharing the local input data, the clients will upload their model parameters to the server periodically. A communication round is used for client upload, server aggregation, and server distribution. Here we introduce the FedAvg (McMahan et al., 2017) communication for client-server update.

- In the $r$-th communication round, each client $i$ uploads it's local model parameters change $\Delta\theta_i^{(r)}$ to the server. The ervser aggregates the local model updates from all participating clients, given by

$$\theta^{(r)} = \theta^{(r-1)} + \frac{1}{K} \sum_{i=1}^{K} \Delta\theta_i^{(r)}, \tag{2}$$

  after which the server distributes the aggregated parameters $(\theta^{(r)})$ to clients.
- Upon receiving the aggregated parameters from server, the clients start their own local learning using the private datasets, $i.e.$, $\{\mathcal{D}_i\}_{i=1}^{K}$, which creates new model parameters update for the next round $\Delta\theta_i^{(r+1)}$, given by

$$\Delta\theta_i^{(r+1)} = \eta_i \nabla_{\theta^{(r)}} \ell_{(\boldsymbol{x}_i, \boldsymbol{y}_i) \sim \mathcal{D}_i}(\boldsymbol{x}_i, \boldsymbol{y}_i, \theta^{(r)}), \tag{3}$$

  where $\eta_i$ is the learning rate in local learning. The clients usually update multiple iterations with gradient descent (we only show one update in above equation).

**Gradient-Inversion Attack.** Proposed in Zhu & Han (2020), the Deep Leakage from Gradients (DLG) attack can utilize the gradient information to reverse the private local data and the label information, that is, given the model parameters update $\Delta\theta_i$ one can obtain the similar $(\boldsymbol{x}_i, \boldsymbol{y}_i)$ pairs. This is done by mimicking the real gradient (also the parameters update, see Eq. (3)). Formally, DLG first randomly initialize a dummy input and a dummy label $(\boldsymbol{x}'_i, \boldsymbol{y}'_i)$, and optimize them by minimizing the discrepancy between real gradients and current gradients, given by

$$\min_{\boldsymbol{x}'_i, \boldsymbol{y}'_i} ||\nabla_\theta \ell(\boldsymbol{x}'_i, \boldsymbol{y}'_i) - \nabla_\theta \ell(\boldsymbol{x}_i, \boldsymbol{y}_i)||^2_F + \alpha R(\boldsymbol{x}'), \tag{4}$$

where $\nabla_\theta \ell(\boldsymbol{x}_i, \boldsymbol{y}_i)$ is the real gradient uploaded to the server. $R(\boldsymbol{x}')$ is an image prior loss function with $\alpha$ as the coefficient. The current gradient $\nabla_\theta \ell(\boldsymbol{x}'_i)$, therefore, can be made as close as possible to the real gradient and generate corresponding input data and label.

So far, some defensive methods have been proposed. For example, Gradient Pruning (Zhu & Han, 2020), Gradient Noise (Zhu & Han, 2020), and Mixup (Zhang et al., 2017) aim to provide less or distributed information in gradients. However, these methods sacrifice task performance for better privacy. In this paper, we seek a method for preserving privacy during federated learning while not jeopardizing its accuracy.

# 4 METHODOLOGY

In this section, we introduce our method—combining both Artificial Neural Networks (ANNs) and Spiking Neural Networks (SNNs) for privacy-preserving federated learning.

## 4.1 SPIKING NEURAL NETWORKS

Compared to artificial neurons *i.e.*, ReLU: $\max(0, x)$, spiking neurons are biologically-inspired, where each neuron maintains a variable dubbed membrane potential $\boldsymbol{v}$. Here, we describe the dynamics using the iterative expression of the Integrate-and-Fire (IF) neuron model (Liu & Wang, 2001), which is favorable for ANN-SNN conversion regime (Rueckauer et al., 2016; Han et al., 2020). Formally, at time step $t$, the IF neuron receives the pre-synaptic input, and then charges the membrane potential, given by

$$\boldsymbol{v}(t) = \boldsymbol{v}(t) + \boldsymbol{I}(t), \quad \boldsymbol{s}(t) = \begin{cases} V_{th} & \text{if } \boldsymbol{v}(t+1) \geq V_{th} \\ 0 & \text{otherwise} \end{cases}, \quad \boldsymbol{v}(t+1) = \boldsymbol{v}(t) - \boldsymbol{s}(t), \tag{5}$$

where $\boldsymbol{I}(t)$ is the pre-synaptic input calculated by the weights $\boldsymbol{W}$ and the spike $\boldsymbol{s}$ from last layer. As long as the membrane potential exceeds the firing threshold $V_{th}$, the neuron will elicit a spike $\boldsymbol{s}$, otherwise, it will stay silent. For fired neurons, the membrane potential will be reset by subtraction, *i.e.*, the third term in Eq. (5). Noting that the threshold $V_{th}^{(l)}$ differs between layers, the server can further apply weight normalization (Diehl et al., 2016) to convert the output $\{0, V_{th}^{(l)}\}$ to a binary spike $\{0, 1\}$.

## 4.2 SYNERGISTIC NEUROMORPHIC FEDERATED LEARNING

In the conversion from a source ANN to a target SNN, the discrepancy of output activation between the two networks will accumulate layer-by-layer, resulting in significantly different output at the final output layer. To address this issue, a layer-wise parameter calibration technique (Li et al., 2021b) is proposed to adjust the SNN parameters so that its activation frequency gets close to the activation in the source ANN. Mathematically, denote the average spike rate over time in SNN is $\bar{\boldsymbol{s}}$, we can write the conversion error as $\boldsymbol{e} = \boldsymbol{a} - \bar{\boldsymbol{s}}$, where $\boldsymbol{a}$ is the activation in source ANN. For each channel $c$, the average activation is then calculated as $\mu_c(\boldsymbol{a}) = \frac{1}{wh}\sum_{i=1}^{w}\sum_{j=1}^{h} x_{c,i,j}$, where $w, h$ are the width and height of the feature. The bias calibration (BC) algorithm computes the spatial mean of the error term, given by

$$\mu_c(\boldsymbol{e}) = \mu_c(\boldsymbol{a}) - \mu_c(\bar{\boldsymbol{s}}). \tag{6}$$

Afterwards, $\mu_c(\boldsymbol{e})$ can be added to the $c$-th channel of bias term $\boldsymbol{b}$ in SNNs. When calculating $\mu_c(\boldsymbol{e})$, we need to estimate $\mu_c(\boldsymbol{a})$ and $\mu_c(\bar{\boldsymbol{s}})$ based on a small calibration dataset (e.g. 128 images), which is not accessible on the server set. Given that the small calibration dataset is a subset of the client's

private dataset and differs between clients, the attacker will be unable to recover every client's $\mu_c(\boldsymbol{e})$. Our framework can take advantage of this algorithm to encrypt the gradient information, therefore improving privacy in FL. The overall pipeline (Algo. 1) in our SNFL is described as follows:

**Client Encryption.** Before uploading the parameter updates to the server, the clients convert the ANN models into SNN models. The conversion first replaces all the ReLU neurons into IF neurons. Due to there is no corresponding module in SNN for BN layer, Rueckauer et al. (2017) propose to absorb the BN parameters to the weight and bias, which can be represented as $W_S \leftarrow W_A \frac{\gamma}{\sigma}$, $b_S \leftarrow \beta + (b_A - \mu)\frac{\gamma}{\sigma}$, where $W_A, b_A$ are the weight and bias of ANN model, $W_S, b_S$ are the weight and bias of SNN model, $\mu, \sigma$ are the running mean and standard deviation and $\gamma, \beta$ are the transformation parameters of the Batch Normalization (BN) layer. Then, all clients run bias calibration algorithm to update the bias parameters for SNN. Note that each local BC process will infer its own local model using its private dataset to record some activation, *i.e.*, $\boldsymbol{a}_i$ and $\bar{\boldsymbol{s}}_i$ for the $i$-th client's ANN and SNN. The bias parameters can then be calibrated as $\boldsymbol{b}_S^i{}' \leftarrow \boldsymbol{b}_S^i + \mu(\boldsymbol{a}_i) - \mu(\bar{\boldsymbol{s}}_i)$. Because the curious server couldn't recover $\mu(\boldsymbol{a}_i) - \mu(\bar{\boldsymbol{s}}_i)$, and that couldn't recover $b_S$ from $b_S'$, it couldn't possibly recover $b_A$. The BC algorithm has two advantages: (1) the uploaded SNN has higher accuracy since its parameters are calibrated, and (2) the uploaded parameters are not the same as the original ANNs, which prevents leakage from gradients. In the sharing step, clients send parameters to the server, which include the weight ($W_S^i$), bias ($b_S^i{}'$), threshold ($V^i$) of SNN models as well as BN layer ($\gamma^i, \sigma^i, \beta^i, \mu^i$) of ANN models.

**Server Aggregation.** In the global communication round, the server will receive the encrypted parameters from clients. Then, on the server side, we apply FedAvg (McMahan et al., 2017) (*cf.* Eq. (2)) to aggregate the massive clients updates. The server averages parameters uploaded by all clients to obtain the averaged SNN model ($W_S^S, b_S^S, V^S$) and BN layer parameters ($\gamma^S, \sigma^S, \beta^S, \mu^S$), where the subscript $S/A$ indicates that the parameters are from the SNN or ANN model and the superscript $S$ indicates that the model is owned by the server. Thus, the aggregated model is also SNN. Since SNN has discrete spikes, it might be less sensitive to a small amount of random noise, which leads to the robustness of SNN (Venkatesha et al., 2021). In practice, we find that the aggregation of SNNs loses less task performance than that of ANNs in most cases. Note that clients do not require the aggregated SNN's parameters. So the server has to take an additional step while processing the $W_A$ from each client to obtain the parameters that clients require. The server gets $W_A^i \leftarrow W_S^i \frac{\sigma^i}{\gamma^i}$ from the $i$-th client and then applies (*cf.* Eq. (2)) to aggregate all $W_A$ from clients to obtain $W_A^S$, which is the parameter required by clients. In the sharing step, the server sends $W_A^S, \gamma^S, \sigma^S, \beta^S, \mu^S$ to each client.

**Server Distribution.** Following the FedAvg setting, clients use these parameters updated by the server to recreate the ANN model and continue to train the ANN using the local dataset. Note that clients will not receive the updates of bias parameters from the server; rather, they continue to use their original ANN bias (before calibration) for the next round of training. On the client-side, they lose less information compared to FedAvg since the bias parameters are kept intact, which helps them learn a better local model. On the server-side, it always uses SNN model for evaluation.

## 5 EXPERIMENTS

In FL, it is frequently vital to consider the presence of semi-honest (honest-but-curious) adversaries for the sake of privacy protection. The adversary is honest in the sense that he/she faithfully follows the collaborative learning protocol, but he/she may be curious about the training data of other participants. On the one hand, given the presence of semi-honest partners, private data must be kept as secure as possible, while a certain amount of information must be transferred across parties for the sake of learning utility. In this section, we conduct experiments to demonstrate the benefits of SNFL in protecting privacy and, in most cases, improving accuracy.

### 5.1 PRIVACY

The parameters that the server can get are $\Delta W_S^i(r) = W_S^i(r) - W_S^i(r-1)$, $\Delta W_A^i(r) = W_A^i(r) - W_A^i(r-1)$, $\Delta b^i(r)' = b^i(r)' - b^i(r-1)'$, where $r$ is the global round and $i$ is the $i$-th client.

---

**Algorithm 1** Synergistic Neuromorphic Federated Learning (SNFL)

---

**Input**:Set of $K$ clients with local datasets; B is the local minibatch size, $E_L$ is the number of local epochs, $E_G$ is the number of Global communication round, and $\eta$ is the learning rate.
**Parameter**: $f_{SG}$ is the global SNN model of server, $f_{SL}$ is the local SNN model of client, $f_{AL}$ is the local ANN model of client
**Output**: Well-trained model $f_{SG}$
**Algorithm**: MAIN

 1: Initialize global SNN model $f_{SG}$ and local ANN model $f_{AL}$ with random weights
 2: **for** round $m \leftarrow 0$ to $E_G$ **do**
 3:     Broadcast the weight parameters of current global ANN model $w' \in f_{SG}$ to all clients
 4:     **for** Client $c \leftarrow 0$ to $K$ **do**
 5:         Transplants $w'$ to local ANN model $f_{AL}$
 6:         **for** epoch $n \leftarrow 0$ to $E_L$ **do**
 7:             Train $f_{AL}$ with local private dataset
 8:         **end for**
 9:         Perform ANN-TO-SNN() and obtain $f_{SL}$
10:     **end for**
11:     Randomly select $P$ participating clients
12:     Aggregate the parameters of $f_{SL}$ using FedAvg and obtain $f_{SG}$
13:     Send $f_{SG}$ parameters to the clients
14: **end for**

**Procedure**:ANN-TO-SNN ()

 1: **for** all $i = 1, 2, \ldots, p$-th layers in the ANN **do**
 2:     Collect input data $x^{(i)}$,output data $x^{(i+1)}$ in one batch
 3:     Get MMSE threshold $V_{th}^{(i)}$ using grid search
 4:     Get SNN output $\bar{s}^{(i+1)}$
 5:     Compute Error term $e^{(i+1)} = x^{(i+1)} - \bar{s}^{(i+1)}$
 6:     Calibrate bias term $b(i) \leftarrow b(i) + \mu(e^{i+1})$
 7: **end for**
 8: **output** Converted SNN model

---

**Attack on SNN model:** We employ gradient inversion attack (Zhu & Han, 2020) (*cf.*) Eq. (4) on LeNet (LeCun et al., 2015) with batch size 1 and optimize for 280 iterations. We use CIFAR10 (Krizhevsky et al., 2010) to evaluate the attack and defense performance. The leaking process is visualized in Fig. 2 (a). This attack can recover every image from ANN model gradients. However, when the model is converted into SNN, the attack is rendered ineffective. This is because, as mentioned in 4.2, SNN replaces the differentiable ReLU neurons into non-differentiable IF neurons. For SNN, there is no gradient.

**Attack on ANN model converted from SNN model:** We also consider a case where an attacker ignores the bias calibration and forcibly converts the SNN model to an ANN model. We employ a more sophisticated gradient inversion attack (Huang et al., 2021) on ResNet20 (Sengupta et al., 2018) with a more realistic setting in which the attacker is unaware of the exact batch size. $ANN1 \rightarrow SNN \rightarrow ANN2$. The difference between $ANN2$ and $ANN1$ is the value of bias. As shown in Fig. 2 (b), the images reconstructed by ANN2 are more blurry than the images reconstructed by ANN1.

## 5.2 ACCURACY

**Implementation Details:** We perform experiments on widely adopted benchmarks CIFAR10 (Krizhevsky et al., 2010), CIFAR100 (Krizhevsky et al., 2010), and Tiny-ImageNet constructed from ImageNet (Russakovsky et al., 2015). To simulate federated learning scenario, we randomly split the training set of each dataset into $N$ parties, and assign one training party to each client. Namely, each client owns its local training set. We are interested in different partitions: IID and Non-IID, where the overall label distribution across clients is the same in the IID setting, whereas class proportions and the number of data points of each client are unbalanced in the Non-IID setting. Especially, for the Non-IID setting, we impose data shift as follows (Li et al., 2022): The size of the local dataset $|D^i|$

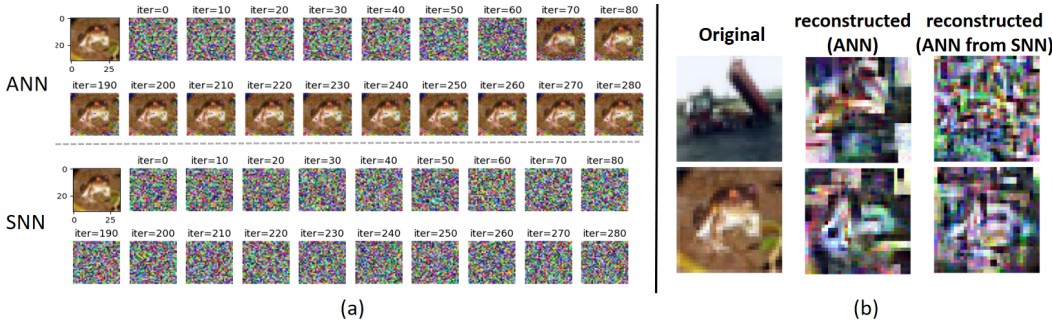

(a)                                                                                          (b)

Figure 2: Results of gradient inversion attack. The left half shows the attack on ANN model and SNN model, respectively. The right half shows the attack on ANN model and ANN model reconverted from SNN model.

Table 1: Accuracy(%) on CIFAR10, CIFAR100, and TinyImageNet test dataset with IID. "P" means the number of participating clients. "Client Accuracy" shows the mean and variance of all clients' ANN models. "Server Accuracy" is the accuracy of the server model."S-" means applying SNFL.

| Dataset | P | FedAvg | S-FedAvg | FedProx | S-Fedprox | MOON | S-MOON | SOLO |
|---|---|---|---|---|---|---|---|---|
| | | | | **Client Accuracy** | | | | |
| CIFAR10 | 5 | 94.4±0.06 | 94.7±0.06 | 94.6±0.06 | 95.4±0.05 | 94.8±0.08 | 95.3±0.04 | 89.6±0.18 |
| | 10 | 92.0±0.11 | 93.1±0.07 | 91.4±0.08 | 95.1±0.09 | 92.9±0.07 | 94.7±0.09 | 84.2±0.89 |
| | 15 | 88.6±0.18 | 91.1±0.08 | 84.8±0.10 | 95.0±0.12 | 89.5±0.09 | 94.4±0.08 | 79.7±1.07 |
| CIFAR100 | 5 | 70.1±0.08 | 71.4±0.22 | 70.3±0.12 | 72.8±0.39 | 71.5±0.24 | 72.2±0.56 | 52.4±0.88 |
| | 10 | 67.0±0.13 | 68.9±0.17 | 67.5±0.19 | 72.8±0.18 | 69.2±0.10 | 72.1±0.45 | 40.2±0.46 |
| | 15 | 60.2±0.26 | 66.8±0.19 | 59.7±0.11 | 71.7±0.22 | 67.2±0.21 | 71.7±0.30 | 31.4±2.50 |
| Tiny-ImageNet | 5 | 39.0±0.02 | 43.7±0.01 | 38.2±0.01 | 41.7±0.00 | 25.2±0.12 | 34.6±3.08 | 19.5±1.45 |
| | 10 | 31.0±0.01 | 41.3±0.03 | 29.9±0.00 | 39.2±0.02 | 21.5±0.38 | 34.9±1.22 | 9.38±0.66 |
| | 15 | 22.1±0.02 | 35.9±0.04 | 26.5±0.02 | 37.7±0.31 | 18.2±0.24 | 32.8±0.84 | 5.31±0.38 |
| | | | | **Server Accuracy** | | | | |
| CIFAR10 | 5 | 94.5 | 95.2 | 94.6 | 95.2 | 94.9 | 94.4 | / |
| | 10 | 92.2 | 95.3 | 91.6 | 95.1 | 92.8 | 94.6 | / |
| | 15 | 88.8 | 94.8 | 85.0 | 95.0 | 89.5 | 94.3 | / |
| CIFAR100 | 5 | 70.2 | 73.1 | 70.4 | 73.5 | 72.2 | 73.6 | / |
| | 10 | 62.1 | 72.3 | 67.8 | 73.5 | 69.6 | 73.0 | / |
| | 15 | 60.9 | 72.4 | 60.1 | 72.0 | 68.0 | 72.2 | / |
| Tiny-ImageNe | 5 | 38.9 | 43.4 | 38.2 | 41.5 | 30.5 | 35.2 | / |
| | 10 | 31.1 | 41.3 | 30.0 | 38.7 | 25.9 | 35.5 | / |
| | 15 | 22.9 | 35.9 | 26.6 | 35.9 | 20.9 | 33.4 | / |

varies across clients. We sample $q \sim Dir_N(\beta)$ and allocate a $q_i$ proportion of the total data samples to client $i$. We should note that if $\beta$ is set to a smaller value, then the partition is more unbalanced. we use $q \sim Dir_N(\beta)$ to denote such a partitioning strategy.

For all the experiments, we use ResNet20 (Sengupta et al., 2018) architecture and SGD optimizer with a weight decay of 1e-5 and momentum of 0.9. The adopted learning is 0.1, which is multiplied by 0.1 at communication rounds 61 and 96. We set the total global communication rounds $E_g$ at 100 and train each client for $E_l = 5$ epochs in every global communication round. The simulation length T of SNN model is 256.

**Baselines:** **(1) FedAvg** (McMahan et al., 2017): it involves multiple local random gradient updates on the client nodes, followed by server model averaging updates. **(2) FedProx**(Li et al., 2020): it improves the local objective based on FedAvg and introduces an additional $L_2$ regularization term in the local objective function to limit the distance between the local model and the global model. A hyper-parameter $\mu$ is introduced to control the weight of the $L_2$ regularization. **(3) MOON** (Li et al., 2021a): it corrects the local updates by maximizing the agreement of representation learned by the current local model and the representation learned by the global model. **(4) SOLO:** each client uses local private data to train its own model without communicating with others.

Table 2: Accuracy(%) on CIFAR10 and CIFAR100 with different degrees of Non-IID. There are 10 participating clients.

| Dataset | Partition | FedAvg | S-FedAvg | FedProx | S-FedProx | MOON | S-MOON |
|---------|-----------|--------|----------|---------|-----------|------|--------|
| CIFAR10 | $q \sim Dir_N(2.0)$ | 92.15 | 94.74 | 92.54 | 94.87 | 92.59 | 94.74 |
| | $q \sim Dir_N(1.0)$ | 90.83 | 94.6 | 91.8 | 94.45 | 91.83 | 94.6 |
| | $q \sim Dir_N(0.5)$ | 88.67 | 86.88 | 89.97 | 93.88 | 90.95 | 93.38 |
| CIFAR100 | $q \sim Dir_N(2.0)$ | 68.81 | 72.54 | 68.52 | 72.4 | 69.07 | 73.95 |
| | $q \sim Dir_N(1.0)$ | 66.3 | 72.47 | 67.58 | 71.78 | 70.27 | 73.02 |
| | $q \sim Dir_N(0.5)$ | 68.17 | 71.47 | 67.81 | 71.63 | 70.23 | 72.29 |

Table 3: The impact of bias. "✓/✗" denotes whether to do bias calibration (BC), or whether the server shares its bias with clients (Bias Back). "T=" means the simulation length T of SNN model. The dataset is CIFAR10.

| BC | Bias back | T=64 | T=128 | T=256 |
|----|-----------|------|-------|-------|
| ✗ | ✗ | 94.46 | 95.08 | 95.14 |
| ✗ | ✓ | 94.4 | 94.82 | 95.00 |
| ✓ | ✗ | **94.75** | **95.05** | **95.14** |
| ✓ | ✓ | 94.51 | 94.91 | 95.08 |

**Validation in the IID Case:** Since a real-world federated system involves many devices, a federated learning model must be scalable with the number of devices. In this experiment, we verify the cases of 5, 10, and 15 clients, respectively. As shown in Table. 5.1, we present the test accuracy on all datasets before and after applying SNFL. From a horizontal perspective, it can be observed that applying SNN-conversion increases accuracy for all baseline methods, even with an accuracy gain of up to 13.79%. This is particularly inspiring because SNPL requires no modification to the original federated training process. One can easily get considerable accuracy profits by simply post-processing the trained global model. Comparing the accuracy gains of different methods after applying SNFL and whole data calibration, we find that FedProx and MOON have the greatest improvement. From the vertical, the accuracy of baseline methods using SNFL drops more slowly as the number of clients increases. For example, on CIFAR100, when 5 clients participate, the accuracy of S-MOON is 0.71% higher than MOON, while the difference between S-MOON and MOON increases to 4.46% when 15 clients participate. This reflects the SNFL benefits that are better suited to the federated learning situation with a large number of clients.

**Validation in the Non-IID Case:** A key challenge in FL is the Non-IID data among the parties. As shown in Table. 5.2, although accuracy decreases to varying degrees as $\beta$ decreases, SNFL can still increase the performance of model, with MOON and Fed seeing the most benefit. For instance, on CIFAR10, when $\beta = 0.5$, S-FedProx is 3.91% higher than FedProx and S-MOON is 2.43% higher than MOON.

## 5.3 BIAS ANALYSIS

Compared to FedAvg, SNFL has two different operations on bias. (1) In the conversion between ANN and SNN, one client uses its local private data to calibrate the SNN model's bias. (2) When sending back server's model to clients, standard FL returns all parameters of server's model including bias, but the server of SNFL keeps bias as a private key and does not share it with clients. Since both clients and server use bias as their private key, SNFL secures the privacy of all parties to some extent. To investigate the effect of the above two operations on bias, we design four ablation studies (Table. 5.2). It can be seen that the extra BC operations have no significant effect on the final server accuracy when clients use their own saved bias instead of the server's bias. There is a slight increase in accuracy if the server does not share its bias with clients. In other words, we are protecting privacy while maintaining the performance of the model, rather than trading precision for privacy as in previous studies.

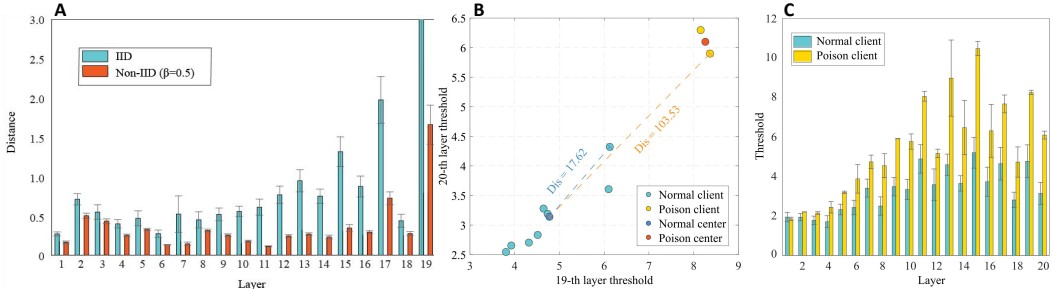

Figure 3: Bias and threshold analysis. *(A)* The L2 distance between the true bias and the inferred bias. *(B)* Scatter image for the thresholds of the last two layers. 2/10 clients are attack clients. *(C)* The thresholds comparison between the normal clients and the poisoned clients. The dataset is CIFAR10.

Then, we simulate an honest but curious server, which attempts to infer one client ANN's full parameters. In this experiment, we used 10 clients with IID and Non-IID ($q \sim Dir_N(0.5)$). Since the malicious server cannot directly access the bias of ANN in SNFL, it can only calculate a new bias using the bias of SNN and ANN's BN parameters. Fig. 3 (A) shows the L2 distance between the true bias and the inferred bias. In the IID case, the disparity between them is much greater than in Non-IID case, showing BC needs more huge bias shift (more huge difference between ANN and SNN) when layer-by-layer calibration in the IID case.

## 5.4    BACKDOOR ATTACK AND DETECTION

In federated learning, backdoor attack (Bagdasaryan et al., 2020) attempts to cause the model to make wrong judgments about data with a certain characteristic (trigger), but the model does not affect on the main task. In other words, the attacked model still exhibits high accuracy on the test dataset, but its output will be different from the output of the clean model when input activates the backdoor trigger. Since the server cannot access the client's training data in federated learning, it is difficult to determine whether the global model has been poisoned through data detection.

In recent paper (Bhagoji et al., 2019), the server identifies the abnormal client based on clustering or mean detection on the model weight and bias. However, these detection methods are cumbersome since the number of parameters in ANN is enormous, e.g., the number of parameters in ResNet20 has reached over 11 million. For SNFL, the server adopts the SNN model, which has a special type of parameter called "threshold". As shown in Fig. 3 B and C, we find that the thresholds (only 20 thresholds in ResNet20, including the converted pooling layer) of poisoned clients and normal clients are significantly different. The distance between the normal clients' threshold set and the cluster center is less than 17.62, whereas the distance between the poisoned client and the cluster center is more than 103.53, making the distinction very clear.

## 6    CONCLUSION

In this paper, we propose that SNFL, a simple FL framework, protects privacy while improving model accuracy in most cases. To the best of our knowledge, this is the first paper that allows clients and server to use different types of neural network models. SNFL can be thought of as a lightweight encryption component add-on for any global federated objective. We investigate the root cause of its privacy from both theoretical and experimental perspectives and conduct experiments to demonstrate that the framework does not jeopardize the model's performance. Empirical results demonstrate that SNFL can result in both more privacy and more accurate models compared with the strong baseline. Our work suggests several interesting directions for future studies, such as exploring the applicability of SNFL to other attacks and its ability to migrate to other algorithms.

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

## A   APPENDIX

### A.1   IMPACT OF NUMBER OF CLIENTS

The total number of clients is one aspect of scalability that influences the performance of a federated learning system. To study the impact of client count, we use the case with CIFAR10 in IID setting. We observe, in Table. A.1, that there is a sharp drop as the number of clients increases in standard FL. However, in SNFL, the server model accuracy lowers just little as show in Fig. A.1.

Table 1: Impact of Number of Clients-CIFAR10

| Dataset | Party | FedAvg | S-FedAvg | FedProx | S-Fedprox | MOON | S-MOON |
|---|---|---|---|---|---|---|---|
| **Clients** | 5 | 94.38±0.06 | 94.71±0.06 | 94.59±0.06 | 95.40±0.05 | 94.80±0.08 | 95.26±0.04 |
| | 10 | 92.02±0.11 | 93.09±0.07 | 91.39±0.08 | 95.11±0.09 | 92.90±0.07 | 94.73±0.09 |
| | 15 | 88.62±0.18 | 91.10±0.08 | 84.76±0.10 | 95.01±0.12 | 89.50±0.10 | 94.36±0.08 |
| | 60 | 57.01±0.18 | 90.14±0.02 | 73.46±0.05 | 90.88±0.02 | 59.68±0.22 | 90.57±0.038 |
| **Server** | 5 | 94.45 | 95.22 | 94.64 | 95.18 | 94.85 | 94.44 |
| | 10 | 92.19 | 95.25 | 91.6 | 95.07 | 92.81 | 94.62 |
| | 15 | 88.80 | 94.76 | 84.97 | 94.98 | 89.5 | 94.27 |
| | 60 | 57.62 | 90.31 | 73.94 | 91.00 | 60.48 | 90.66 |

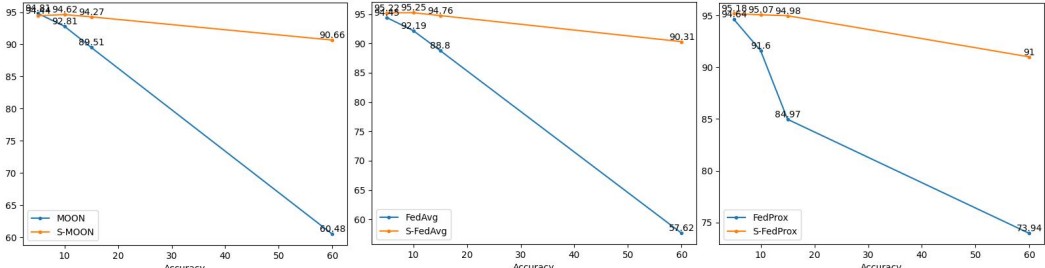

Figure 1: Impact of Number of Clients-CIFAR10

### A.2   SENSITIVITY TO STRAGGLERS

Because real-world networks are inherently unstable, assuming updates communication would be effective from all selected devices is impracticable. Hence, the model needs to be robust enough to handle devices that fail to communicate updates. These devices are referred to as stragglers. In this section, we analyze the impact of stragglers on the performance of final SNN model. We use a case with a total of $N = 60$ clients. In each round, we randomly select $PN$ clients upload parameters to server. We consider different levels of probabilities and summarize in Table. A.2. We found that when the number of stragglers decreased, the accuracy of FedAvg increased first and subsequently declined to 57.62%, whereas the accuracy of S-FedAvg increased amazingly consistently to 90.31%.

Table 2: The accuracy with various client drop probabilities.

| Probability | 0.1 | 0.2 | 0.5 | 0.7 | 1 |
|---|---|---|---|---|---|
| FedAvg | 65.07 | 66.05 | 68.16 | 65.76 | 57.62 |
| S-FedAvg | 73.29 | 83.53 | 89.87 | 90.13 | 90.31 |

### A.3   SENSITIVITY TO THE LENGTH OF SIMULATION T

The forwarding pass in SNN is repeated for $T$ steps to get the final result, where the final result is the expectation of the ultimate layer's output across $T$ steps. This allows the flexibility of adjusting $T$ to

balance between the latency and accuracy of SNNs for different application scenarios. We conduct experiments on CIFAR10 with different simulation length $T$, as shown in Table. A.3. We discovered that increasing T improved SNN accuracy to a certain extent. However, as $T$ increases to a certain point, its influence on accuracy decreases.

Table 3: The accuracy with different SNN simulation steps.

| Simulation steps | 64 | 128 | 192 | 256 | 320 |
|---|---|---|---|---|---|
| S-FedAvg | 89.26 | 90 | 90.23 | 90.31 | 90.26 |
| S-FedProx | 90.52 | 90.96 | 90.94 | 91 | 90.99 |
| S-MOON | 89.95 | 90.51 | 90.66 | 90.66 | 90.74 |

