# OpenReview forum: "Synergistic Neuromorphic Federated Learning with ANN-SNN Conversion For Privacy Protection"
_ICLR.cc/2023/Conference — Submitted to ICLR 2023_

### Official Review · Reviewer_Wfez · 2022-10-13

**Confidence:** 4
**Correctness:** 3
**Technical Novelty And Significance:** 2
**Empirical Novelty And Significance:** 2
**Recommendation:** 3

**Clarity, Quality, Novelty And Reproducibility:**

Clarity: 5/10

Quality: 5/10

Novelty: 4/10

Reproducibility: 5/10

**Strength And Weaknesses:**

Strengths:
1. The tackled problem is relevant to the ICLR community.
2. The results look promising and competitive.

Weaknesses:
1. There are several typos throughout the text. Please revise and correct.
2. The proposed method combines multiple techniques together. Besides such a combination, each individual technique seems strongly influenced by previous works. It is recommended to discuss in more detail the key differences between the related works and the proposed method.
3. The table numbering in the text does not match the numbers in the table captions.
4. Please specify which dataset has been used for generating the results reported in Table 3 and Figure 3.
5. It would be useful to provide the source code for reviewers' inspection during the rebuttal.

**Summary Of The Paper:**

A framework for private federated learning is proposed. The conversion from ANN to SNN includes the encryption property for privacy protection. The results show competitive performance for IID and non-IID cases.

**Summary Of The Review:**

The idea of the paper is interesting but several concerns should be clarified.

---

> ### Author Response · Authors · 2022-11-16
> **Re: Official Review of Paper1560 by Reviewer Wfez**
>
> Many thanks for your valuable time to review our paper. We are glad that you acknowledge that the problem is interesting and the framework we propose performs well. We will elaborately address your concerns and questions as follows.
>
> >**W**: The proposed method combines multiple techniques together. Besides such a combination, each individual technique seems strongly influenced by previous works. It is recommended to discuss in more detail the key differences between the related works and the proposed method.
>
> **A**:  When reading some papers on privacy protection, we have been thinking about the intrinsic source of privacy in these defense strategies. If we consider this question from an information theory perspective, there is indeed an asymmetry of entropy in the encryption and decryption steps for clients and servers when partial encryption information is kept locally only. Adding random noise to the model is one of the easiest ways to generate entropy asymmetry. As we all know, the more noise there is, the less accurate the model is. So is there a type of beneficial noise that does not harm the model's accuracy but actually improves it? This is satisfied by the deviation generated by the model conversion. The noise, on the other hand, works because it is so random that the server has no idea what it is and cannot obtain the model before the noise is added. Similarly, we must include a value in conversion that the server cannot obtain. This is satisfied by the deviation generated by the model calibration. The ANN-SNN conversion and calibration are exactly what we require. This is why we want to combine these technologies. Until we wrote this paper, no one had considered privacy in federated learning from this angular position. We believe that thought, not technology, is the source of novelty, and we hope that our ideas will inspire other researchers.
>
> >**W**: Please specify which dataset has been used for generating the results reported in Table 3 and Figure 3.
>
> **A**: CIFAR10 has been used for these experiments in Table 3 and Figure 3. We have explained this in the revised version.
>
> >**W**: There are several typos throughout the text. Please revise and correct. The table numbering in the text does not match the numbers in the table captions.
>
> **A**: Thanks for the correction. We have fixed this in the revised version.
>
> >**W**: It would be useful to provide the source code for reviewers' inspection during the rebuttal.
>
> **A**: Yes, we have provided the source code.

---

> > ### Comment · Reviewer_Wfez · 2022-11-21
> > **Response to Authors**
> >
> > The efforts made by the authors in answering the reviewers' comments are appreciated.

---

> > > ### Author Response · Authors · 2022-11-21
> > > **Would you mind re-evaluating our work?**
> > >
> > > Thanks for the recognition of our effort. Although our initial score is not optimistic, we are confident in the quality of our work. Our contribution is NOT a simple combination of several existing works but integrating the SNN and ANN into an asymmetric encryption paradigm for federated learning. As we explain in the rebuttal, this is a novel idea that has not been identified in the field. We believe it is worth sharing with the community. Thus we sincerely hope you can check our rebuttal and re-evaluate our work. Thank you for your time and patience.

---

### Official Review · Reviewer_55hw · 2022-10-24

**Confidence:** 4
**Correctness:** 3
**Technical Novelty And Significance:** 2
**Empirical Novelty And Significance:** 3
**Recommendation:** 3

**Clarity, Quality, Novelty And Reproducibility:**

This paper has some merits in certain aspects since the ANN-SNN conversion is novel in federated learning. However, the authors did not support their claims well and some extremely relevant related works are not well discussed and compared, which weakens the novelty and solidity of this paper. Apart from the technical parts, the paper is somewhat poor in presentation.

**Strength And Weaknesses:**

Strength:
This paper is somewhat novel, as it provides a solution to break the privacy-utility tradeoff.

Weakness:
- 1 One major weakness is that the most relevant related works are not well discussed and compared, which weakens the novelty and solidity of this paper. Before this submission, there are existing works about spiking neural networks in federated learning and privacy-preserving spiking neural networks, such as
①	Venkatesha Y, Kim Y, Tassiulas L, et al. Federated learning with spiking neural networks[J]. IEEE Transactions on Signal Processing, 2021, 69: 6183-6194.
②	Yang H, Lam K Y, Xiao L, et al. Lead federated neuromorphic learning for wireless edge artificial intelligence[J]. Nature communications, 2022, 13(1): 1-12.
③	Kim Y, Venkatesha Y, Panda P. PrivateSNN: Privacy-Preserving Spiking Neural Networks[C]//Proceedings of the AAAI Conference on Artificial Intelligence. 2022, 36(1): 1192-1200.

The authors did not compare the algorithms about directly incorporating SNN in FL and discuss the accuracy gains and privacy protection in this case. It is likely that the accuracy gains of SNFL are caused by the SNN itself, not the ANN-SNN conversion. Additionally, the author claims that the privacy guarantee of SNFL lies in “the server of SNFL keeps bias as a private key and does not share it with clients”. But in the vanilla FL with SNN, one can keep the bias of SNN as private keys in the server and clients, and it also can realize better privacy protection, so what is the difference and novelty of SNFL when compared with this case?

Besides, the author did not compare the privacy-preserving SNN directly adopting in FL, and they did not discuss the existing works about privacy issues in SNN. As in “Kim Y, Venkatesha Y, Panda P. PrivateSNN: Privacy-Preserving Spiking Neural Networks[C]//Proceedings of the AAAI Conference on Artificial Intelligence. 2022, 36(1): 1192-1200.”, they mentioned the privacy leakages in ANN-SNN conversion. Since the SNFL is just a simple adaptation of ANN-SNN conversion in FL, will these leakages happen in SNFL?

- 2 Another major concern of this paper is that the authors did not well support their claims in certain aspects. Specifically, the authors mentioned “SNFL introduces a lightweight overhead” in the abstract, but the computation overhead of the conversion is not discussed in the main paper. And I reckon the overhead is of particular importance, because the conversion process is conducted at the clients’ sides. While the clients’ devices are always resource-constrained, such a computation of conversion may cause extra burden to the clients. The authors should fully discuss the overhead and compare it with other privacy-preserving methods to show its superiority. By the way, I am not sure whether SNN is lighter than ANN in parameter space, and if SNN introduces more parameters, it will also cause extra communication overhead, which should to be claimed or discussed.

The main point of this paper is that SNFL can provide better privacy protection. However, I find the experimental part about privacy is weak and short. It seems that the author focused more on the accuracy rather than privacy in the experiments.

- 3 The authors need give more insights on their method. Such as, why SNFL works better when the number of clients is larger. And for the privacy parts, I reckon it is not direct for the readers to get why your framework can better protect privacy. More empirical findings, like visualization and case study, or more theoretical findings are needed.

- 4 Apart from the technical parts, the paper is somewhat poor in presentation. As I can see, there are many typos, including wrong spelling of words, incomplete formula symbols, wrong method names, mistakes in citations of tables, and so on. Also, some sentences are also confusing, and I am always lost in what the authors are trying to convey. For example, in the first paragraph of Section 4.2, it says “Considering that μc(e) is dependent on the data distribution rather than distribution free, we cannot use random dataset on the server side for the recovery. Our framework can take advantage of this algorithm to encrypt the gradient information, therefore improving the privacy in FL.”. I don’t see the logic behind the two sentences, and it is confusing why the authors mentioned the calibration dataset on the server. Did they use the calibration dataset on server or not? The authors should correct all these typos in the revision and present better in writing.



**Summary Of The Paper:**

This paper proposes a privacy-preserving framework (SNFL) in federated learning by incorporating the conversion from artificial neural networks to spiking neural networks. SNFL is validated to robust to gradient inversion attack and backdoor attack and it guarantees privacy while improving accuracy.

**Summary Of The Review:**

Overall, this paper doesn’t meet the criteria of publication in ICLR. The major concern is that the authors did not support their claims well and some extremely relevant related works are not well discussed and compared, which weakens the novelty and solidity of this paper. I recommend to reject it and hope the authors can do better in the future version.

---

> ### Author Response · Authors · 2022-11-16
> **Re: Official Review of Paper1560 by Reviewer 55hw**
>
> Thank you very much for your review and affirmation of our work. We'll answer your questions one by one in the following, including some misunderstandings and some essential academic questions worth exploring. We are also very honored to share some of our understandings with you.
>
> >**Q**: One major weakness is that the most relevant related works are not well discussed and compared.
>
> **A**: Here we discuss why we did not directly compare the three papers mentioned by the reviewer.
>
> For [1], we appreciate this paper's effort demonstrating that SNN works in FL, and we cite it in our paper. The main difference between this paper and ours in terms of training is that this one is about direct training, whereas we use the ANN-SNN conversion method. The author uses BNTT to train SNN directly, which requires a long training time for the client. And their final accuracy is much worse than the ANN baseline. Actually, this is one of our paper's advantages: it manages to obtain a highly precise SNN model with almost the same training cost than the ANN-only case. Since the purpose of [1] is neither to protect privacy nor to improve the accuracy of the model, it is unfair to compare accuracy considering two very different training methods. (Currently, the accuracy of an SNN model obtained through direct training is lower than that of an SNN converted from an ANN model.) It is irrelevant to our paper's primary purpose of privacy if we directly train SNN models. Because only SNN, like only ANN, does not create an entropy asymmetry between clients and server.
>
> For [2], the main content of this paper is how to select one device with high capability as a leader in decentralized FL, while the main motivation of our paper is to protect the privacy of clients in centralized federated learning. But what's exciting is that we believe the two papers can be combined, increasing not only SNN training speed but also privacy in decentralized FL scenario.  In the future, we will research this topic, but it cannot be finished during the rebuttal phase due to the heavy workload.
> Hope for understanding.
>
> In [3], they aimed to build SNN from a pre-trained ANN model without leaking sensitive information contained in a dataset. In FL, because the client completes the conversion from ANN to SNN independently, the conversion will not result in the client's data leakage. The data that the client sends to the server is what really causes privacy breaches. The motivation of SNFL is considering is how to maintain model accuracy while protecting privacy. It appears to be a good idea if privacy can be protected during the conversion step. But here are two things we have to think about when implant PrivateSNN in federated learning: (1) The parameters of the Encrypted SNN model are not available to restoring ANN parameters, so it's hard to rebuild client ANN for continue training. (2) As stated in [3], PrivateSNN=DC+SET+OD. The DC step that is aimed at generating synthetic images from a pretrained ANN will cost a huge amount of time and reduce the training speed. Therefore, how to combine the two methods needs further research.
>
> >**Q**: The author claims that the privacy guarantee of SNFL lies in “the server of SNFL keeps bias as a private key and does not share it with clients”. But in the vanilla FL with SNN, one can keep the bias of SNN as private keys in the server and clients, and it also can realize better privacy protection, so what is the difference and novelty of SNFL when compared with this case?
>
> **A**: SNFL's privacy mainly comes from keeping bias. Keeping bias as private is better for privacy because the less information is transmitted, the better. The server is able to do this because clients do not need SNN bias parameters and can train available ANN's bias with local data. However, in the vanilla FL situation, if clients don't upload bias to server, the server will get a broken model. The server can get bias neither from the clients nor from data.
>
> >**Q**: mentioned the privacy leakages in ANN-SNN conversion. Since the SNFL is just a simple adaptation of ANN-SNN conversion in FL, will these leakages happen in SNFL?
>
> **A**: Our assumptions are distinct from PrivateSNN's. In PrivateSNN, the conversion phase is inherently unreliable, meaning that an unreliable node for conversion step will obtain some information from the private dataset. But in SNFL, the conversion step is performed by the client itself. Like in standard FL, we need to prevent information leakage that occurs in parameter transmission.

---

> > ### Author Response · Authors · 2022-11-16
> > **Re: Official Review of Paper1560 by Reviewer 55hw**
> >
> > >**Q**: The authors mentioned “SNFL introduces a lightweight overhead” in the abstract, but the computation overhead of the conversion is not discussed in the main paper.
> >
> > **A**:  Aside from the two steps of ANN-SNN conversion and bias calibration, SNFL has nothing special. Conversion will not produce additional inference time.  As in [4], they stated that the time needed for each bias calibration of VGG16 is 0.098±0.003 and that of MobileNet is 2.29±0.005. In terms of memory, calibrating the bias of ResNet-34 requires only 0.3653MB. All experiments were tested on a single NVIDIA GTX 1080TI with 5 runs on ImageNet.
> >
> > >**Q**: Whether SNN is lighter than ANN in parameter space, and if SNN introduces more parameters, it will also cause extra communication overhead, which should to be claimed or discussed.
> >
> > **A**: Except for a "threshold" parameter of each layer, the converted SNN model has the same number of parameters as the ANN model. There are only 20 threshold parameters for RESNET20, which is a small number in comparison to the total of 11.2M parameters.
> >
> > >**Q**: The main point of this paper is that SNFL can provide better privacy protection. However, I find the experimental part about privacy is weak and short.
> >
> > **A**: We have visually shown this point in Figure 1 of the revised version and highlighted it in red. We've also added new experiment to show the impact of privacy protection. There are two scenarios: (1) Directly reconstruct pictures from SNN; (2) Reconverting SNN to ANN. We've done experiments for both scenarios. Since bias calibration is the most important operation, we calculated and visualized the L2 distance between the true bias and the inferred bias in Figure 3.
> >
> > >**Q**: Why SNFL works better when the number of clients is larger.  It is likely that the accuracy gains of SNFL are caused by the SNN itself, not the ANN-SNN conversion.
> >
> > **A**: One interesting discovery is that in FedAvg (only ANN), if the client does not accept bias parameters from the server, the accuracy of the server model is greatly increased with CIFAR10.
> >
> > Bias has a significant impact on the model, and we are exploring the specific reasons for this. SNFL takes more advantage of it because SNFL clients care less about whether bias is returned than FedAvg clients. And SNFL employs the bias private key to prevent information leaking.
> >
> > |60 clients | Accuracy|
> > | :---------------: | :-------------: |
> > |× Bias back  | 72.15 |
> > |√ Bias back | 57.62 |
> >
> > ----------
> > [1] “Venkatesha Y, Kim Y, Tassiulas L, et al. Federated learning with spiking neural networks[J]. IEEE Transactions on Signal Processing, 2021, 69: 6183-6194.”
> >
> > [2] Yang H, Lam K Y, Xiao L, et al. Lead federated neuromorphic learning for wireless edge artificial intelligence[J]. Nature communications, 2022, 13(1): 1-12.
> >
> > [3] Kim Y, Venkatesha Y, Panda P. Privatesnn: privacy-preserving spiking neural networks[C]//Proceedings of the AAAI Conference on Artificial Intelligence. 2022, 36(1): 1192-1200.
> >
> > [4] Li Y, Deng S, Dong X, et al. A free lunch from ANN: Towards efficient, accurate spiking neural networks calibration[C]//International Conference on Machine Learning. PMLR, 2021: 6316-6325.

---

> > ### Comment · Reviewer_55hw · 2022-11-23
> > **feedback**
> >
> > Dear Authors,
> >
> > I have carefully read your rebuttal and I appreciate your efforts in answering my questions. However, I find most of the problems are not well addressed in the revision, as far as I concern, the time is sufficient to give more detailed experiments and results in the revision to support the key claims more strongly.
> >
> > I suppose the reasons that why the related works are not compared are not totally convincing. Additionally, I see the key insights and novelty of this paper is about the "bias" of the ANN models, and the authors need to study deeper towards this point, but I am disappointed that it was not shown in the revision.
> >
> > Thus, I suggest the authors to have a stronger version of this paper in the future. I appreciate all the efforts the authors made, but unfortunately, I choose to withhold my initial score, hoping the authors can understand. I wish the decision will not bother you.
> >
> > Best wishes,
> > Reviewer 55hw

---

### Official Review · Reviewer_ddxi · 2022-10-25

**Confidence:** 3
**Correctness:** 3
**Technical Novelty And Significance:** 3
**Empirical Novelty And Significance:** 3
**Recommendation:** 6

**Clarity, Quality, Novelty And Reproducibility:**

I am stuck with the very basic premise of the paper as given above, but other than that the paper is clearly presented. The method is properly motivated, and the work is original given that my question can be resolved.

**Strength And Weaknesses:**

Strengths: timely topic (enhancing robustness of FL under privacy attacks) and interesting solution.

Weaknesses: I am confused with the very basic premise here. Each client uploads the updated SNN weights and the BN parameters. The server then converts them to the corresponding local ANN weights before aggregation. So if the eavesdropper can access to the uploaded parameters of a client, then she can reproduce the ANN model weights of that client (as the serve does it easily). So, where's the protection against privacy threats that the ANN-SNN conversion process is supposed to bring to the table?

**Summary Of The Paper:**

Federated learning based on client-side uploading of spiking neural networks (SNNs) is considered. Each client's SNN model parameters are obtained by converting locally updated regular NNs (called ANNs here). The server obtains and downloads the aggregated global ANN parameters from the SNN parameters uploaded from the clients. Experiments show performance improvements and enhanced robustness against attacks, compared to baselines.

**Summary Of The Review:**

The paper deals with a timely topic (enhancing robustness of FL under privacy attacks) and appears to provides an interesting solution. However, I am confused with the very basic premise here. Each client uploads the updated SNN weights (converted from the updated ANN) along with the BN parameters. The server then converts them to the corresponding local ANN weights before aggregation and downloading. So if the eavesdropper can access the uploaded parameters of a client, then she can reproduce the ANN model weights of that client in a given iteration (as the serve does it easily). So, where's the protection against privacy threats that the ANN-SNN conversion process is supposed to bring to the table? My evaluation score would be a tentative one and I will reevaluate after/once my confusion gets cleared.

---

> ### Author Response · Authors · 2022-11-16
> **Re: Official Review of Paper1560 by Reviewer ddxi**
>
> Thank you for your thoughtful comments and positive feedback on our work. Below, we address the concern of the reviewer.
>
> > Q: So if the eavesdropper can access to the uploaded parameters of a client, then she can reproduce the ANN model weights of that client (as the serve does it easily). So, where's the protection against privacy threats that the ANN-SNN conversion process is supposed to bring to the table?
>
>
> **A**: Following the conversion of ANN to SNN, each client runs a calibration algorithm to update the bias for SNN, which can be described as
> $$
> b_S^{\prime} \leftarrow b_S + \mu(e)
> $$
> $$
> \mu(e)=\mu(a) - \mu(\bar{s})
> $$
>
> When calculating $\mu(e)$, we need to estimate $\mu(a)$ and $\mu(\bar{s})$ based on a small calibration dataset (e.g., 128 images), which is confidential and not accessible on the server set. That is to say, $\mu(e)$ is private that provides privacy. Despite the fact that the ANN model's parameters can be calculated using the SNN model, which is directly converted from the ANN model, eavesdropper is unable to precisely obtain the ANN model from the calibrated SNN model without the value of $\mu(e)$.
> $$
> b_S^{\prime} \nrightarrow b_S \rightarrow b_A\nonumber
> $$
>
> Calibration for bias is similar to adding noise to bias, with the difference that adding real noise reduces model accuracy, whereas calibration maintains or even increases model accuracy.
>
> We have visually shown this point in Figure 1 of the revised version and highlighted it in red. We've also added new experiment to show the impact of privacy protection in section 5.1.

---

> ### Comment · Reviewer_ddxi · 2022-11-27
> **thanks for the response**
>
> I thank the authors for the clarification. My concerns are lifted to a certain extent but as the other reviewer pointed out, I feel this paper's core novelty lies in the bias and its use for strengthening privacy protection when employing SNN/FL. Yet the bias analysis provided in section 5.3 is rather weak. Fig.2b for one thing is helpful but not enough to establish improved privacy protection. Having said that, I think this work represents a meaningful contribution to the area of ML/FL. Overall, I raise my score to 6.

---

### Decision · Program_Chairs · 2023-01-20

**Decision:**

Reject

**Justification For Why Not Higher Score:**

- The proposed method combines multiple existing techniques. The novelty is quite limited, and the advantages and contributions are not clear.
- The current version is not well discussed and compared, which weakens the novelty and solidity of this paper.

**Justification For Why Not Lower Score:**

N/A

**Metareview: Summary, Strengths And Weaknesses:**

This paper proposes a privacy-preserving framework (SNFL) in federated learning by incorporating the conversion from artificial neural networks to spiking neural networks. The reviewers appreciate the efforts of the authors. However, they found that most of the problems are not well addressed in the revision. The reviewer suggests to study deeper and do more convincing comparison with related work. The current version is not well discussed and compared, which weakens the novelty and solidity of this paper. Moreover, the proposed method combines multiple existing techniques. The novelty is quite limited, and the advantages and contributions are not clear.

Please take the comments and suggestions from the reviewers in their detail reviews to improve the manuscript for the future submission.